# Proinflammatory Polyphosphate Increases in Plasma of Obese Children with Insulin Resistance and Adults with Severe Type 2 Diabetes

**DOI:** 10.3390/nu14214601

**Published:** 2022-11-01

**Authors:** Marcela Montilla, Andrea Liberato, Pablo Ruiz-Ocaña, Ana Sáez-Benito, Manuel Aguilar-Diosdado, Alfonso Maria Lechuga-Sancho, Felix A. Ruiz

**Affiliations:** 1Research Unit, Hospital Universitario Puerta del Mar, 11009 Cadiz, Spain; 2Instituto de Investigación e Innovación Biomédica de Cádiz (INiBICA), 11009 Cadiz, Spain; 3Medical School, Universidad Cooperativa de Colombia, Villavicencio 500003, Colombia; 4Pediatric Endocrinology and Diabetes, Department of Pediatrics, Hospital Universitario Puerta del Mar, 11009 Cadiz, Spain; 5Clinical Analysis Department, Hospital Universitario Puerta del Mar, 11009 Cadiz, Spain; 6Endocrinology and Metabolism Department, Hospital Universitario Puerta del Mar, and Universidad de Cádiz, 11009 Cadiz, Spain; 7Area of Pediatrics, Medical School, Universidad de Cádiz, 11003 Cadiz, Spain; 8Area of Nutrition and Bromatology, Medical School, Universidad de Cádiz, 11003 Cadiz, Spain

**Keywords:** biomarker, children, inflammation, insulin resistance, obesity, polyphosphate, type 2 diabetes

## Abstract

Obesity increases the risk of insulin resistance and type 2 diabetes through increased inflammation at cellular and tissue levels. Therefore, study of the molecular elements involved in obesity-related inflammation may contribute to preventing and controlling it. Inorganic polyphosphate is a natural phosphate polymer that has recently been attracting more attention for its role in inflammation and hemostasis processes. Polyphosphates are one of the main constituents of human platelets, which are secreted after platelet activation. Among other roles, they interact with multiple proteins of the coagulation cascade, trigger bradykinin release, and inhibit the complement system. Despite its importance, determinations of polyphosphate levels in blood plasma had been elusive until recently, when we developed a method to detect these levels precisely. Here, we perform cross sectional studies to evaluate plasma polyphosphate in: 25 children, most of them with obesity and overweight, and 20 adults, half of them with severe type 2 diabetes. Our results show that polyphosphate increases, in a significant manner, in children with insulin resistance and in type 2 diabetes patients. As we demonstrated before that polyphosphate decreases in healthy overweight individuals, these results suggest that this polymer could be an inflammation biomarker in the metabolic disease onset before diabetes.

## 1. Introduction

One of the biggest current public health problems is obesity, due to its very high incidence that is increasing in all strata of the population, as well as the high morbidity and mortality associated with its health complications [1]. Obesity increases the risk of insulin resistance, a condition in which the body produces insulin but does not use it effectively, producing a reduced glucose uptake [2]. Children and adolescents are especially susceptible to developing insulin resistance [3].

Insulin resistance has been identified as a key trigger of type 2 diabetes in both adults and children [4,5]. Type 2 diabetes, historically an adult disease, is also rising in childhood because of the increasing unhealthy lifestyle habits [6].

Inflammation is a main component in the pathogenesis of insulin resistance and type 2 diabetes at different levels [7,8]. A potential player in inflammatory processes, which has recently been attracting more attention, is inorganic polyphosphate (polyP) [9]. PolyP consists of linear phosphate polymers linked by high-energy bonds; these moieties have been found to be present in all cells and tissues studied [10].

In 2004, we reported that polyP is present in the dense granules of human platelets and that it is secreted upon platelet activation [11]. Since then, other groups have found that polyP modulates the coagulation cascade through its interaction with multiple proteins [12]. PolyP also modulates inflammation by triggering bradykinin release [13], by inhibiting the complement system [14], and by activating endothelial cells to increase adhesion molecule expression and von Willebrand factor secretion [15]. In addition to platelets, polyP has been described as present in elements relevant in inflammation such as mast cell [16], lymphocytes [17], and neutrophil extracellular traps [18], among others.

Despite its importance in hemostasis and inflammation, actual levels of polyP in plasma had not been determined until very recently [19]. In that work, performed in healthy individuals, we found a decrease in polyP content in line with the individual’s body mass index (BMI) [19].

In the work reported here, we studied plasma polyP in children that were overweight or obese. We find that the polymer increases in relation to insulin resistance. We also found that plasma polyP increases in adult patients with type 2 diabetes. In accordance with our results, we discuss the possible relevance of polyP as a biomarker in the definition of “metabolic health” in obesity.

## 2. Materials and Methods

### 2.1. Subjects

#### 2.1.1. Children with Normal Weight, Overweight, and Obesity

Participating children with obesity or overweight were recruited from the Pediatric Endocrinology Outpatient Clinic of the Puerta del Mar University Hospital, where their primary care providers had referred them due to their weight excess. Only children in which organic, syndromic, and monogenic obesity was ruled out were finally included in the study: 10 individuals with BMI > 25 (less than 30), and 10 individuals with BMI > 30 were included. In addition, five children participated as individuals with BMI < 25 when blood tests for other non-acute medical conditions (i.e., pre-anesthesia evaluation) were required. Blood was collected by trained nurses in Vacutainer tubes with sodium citrate. Hemolyzed plasma or plasma containing red blood cells was not used. Table 1 summarizes the anthropometric and biochemical characteristics of this sample.

#### 2.1.2. Patients with Severe Type 2 Diabetes and Controls

Type 2 diabetes adults reported in this study were patients of the Endocrinology Unit at the Puerta del Mar University Hospital (*n* = 10, seven males and three females). Inclusion criteria were diagnoses of long-term diabetes with insulin dependence and complications, such as diabetic foot or ischemic heart disease. The average age of the sample was 66 ± 11 years. Mean fasting plasma glucose levels were 212± 83 mg/dL.

In contrast, the controls were 10 healthy volunteers, recruited in our Research Unit at the Puerta del Mar University Hospital. The exclusion criteria for the control individuals were any medical history of Diabetes or Metabolic disorders. Inclusion criteria for the controls were matched by age and sex with the diabetic patients.

Blood was collected by trained nurses in Vacutainer tubes with sodium citrate. Hemolyzed plasma or plasma containing red blood cells was not used.

### 2.2. Anthropomorphic and Biochemical Parameters

Anthropometric data were measured and evaluated by endocrinologists of the Department of Pediatrics and the Endocrinology and Metabolism Department of the Puerta del Mar University Hospital, as previously described [20].

Biochemical analysis was performed in serum or plasma by standard clinical assays in the Clinical Analysis Department at Puerta del Mar University Hospital. Glucose, total cholesterol (TC), high-density lipoprotein (HDL) and low-density lipoprotein (LDL) cholesterol, triglycerides (TG), insulin, aspartate transaminase activity (GOT), alanine transaminase activity (GPT), and z-scores were measured as described before [20]. Insulin resistance was defined as before [20], according to the ADA criteria: a HOMA-IR above 3.5, fasting insulin > 15 µUI/mL, insulin at 120′ of OGTT > 75 µUI/mL, or insulin at any time point of the curve > 150 µUI/mL.

### 2.3. Plasma and Platelet Polyphosphate Determinations

We used a recently developed assay to determine plasma polyP, harnessing the accumulation of this polymer in the cryoprecipitated fraction of the plasma [19]. Briefly, plasma cryoprecipitates were obtained by centrifugation of frozen plasmas at −80 °C for 24 h and then thawed slowly at 4 °C. Thereafter, samples were resuspended in saline solution and treated with perchloric acid to eliminate proteins [19].

Platelets were isolated from whole blood as described before [21] and extracted with perchloric acid for intraplatelet polyP measurement [21].

Precise polyP contents in acidic extracts of cryoprecipitated plasma and platelets were determined from the amount of phosphate residues (P_i_) released upon treatment with an excess of purified recombinant exopolyphosphatase from *Saccharomyces cerevisiae* (scPPX), as described before [11]. scPPX is a highly specific and active enzyme against polyP [22]. For all experiments described here, polyP concentrations are expressed in terms of their P_i_.

### 2.4. Plasma von Willebrand Factor

Plasma von Willebrand factor antigen (vWF:Ag) was measured using the von Willebrand Factor ELISA Kit (Cat. No 5290) from Helena Laboratories (Beaumont, TX, USA), following the instructions of the manufacturer.

### 2.5. Statistical Analysis

The data obtained were processed using SPSS version 15 statistical software. The Shapiro–Wilks statistical significance method was used in order to test data for normal distribution. Bivariate associations between biochemical analytes and polyphosphate levels were assessed by Sperman correlation analysis. Bivariate analysis was performed by simple linear regression. Multivariate analysis was conducted by multiple linear regression. The Mann–Whitney test was used for the analysis of qualitative vs. quantitative variables. A value of *p* < 0.05 was considered statistically significant.

## 3. Results

### 3.1. Plasma Polyphosphate in Children with Overweight/obesity

First, we evaluated 25 children with the morphological and biochemical characteristics shown in Table 1. We divided them in three groups according to their BMI. In the biochemical parameters of the three groups, there were no significant differences except for insulin, which was clearly higher in the overweight and obese groups. This was reflected in the difference in the HOMA-IR index, an indicator of insulin resistance, which was also significantly higher in the overweight and obese groups. Overall, 80% of the overweight and obese children in the sample presented insulin resistance as measured by HOMA-IR.

Measurement of polyP in the plasma of these children showed clear differences between those with BMI < 25 and those with BMI > 25 and BMI < 30 (Figure 1a). No significant differences in plasma polyP content were found between the BMI > 25 and BMI < 30 groups (Appendix A). Levels of plasma von Willebrand factor, a marker of inflammation that increases in insulin resistance and diabetes [23], were also determined (Figure 1b). Von Willebrand factor levels were higher in the BMI > 25 group (Figure 1b). Moreover, we found a significant correlation between plasma levels of polyP and von Willebrand factor (Figure 1c). In addition, as children with overweight and obesity are defined by percentiles, we calculated the z-scores of all individuals and analyzed their correlation with plasma polyP levels (Appendix A).

Next, plasma polyP data were grouped in relation to insulin resistance (Figure 2a). In the sample studied, 80% of individuals with overweight and obesity presented insulin resistance. Children without insulin resistance presented lower plasma polyP levels (Figure 2a). Likewise, the study of the possible correlations between polyP and all the biochemical parameters in Table 1 resulted in the finding that polyP is significantly related only to insulin level (Figure 2b) and to the activity of the liver enzyme aspartate transaminase (GPT) (Figure 2c).

In addition, in order to control potential confounding variables, multiple linear regression analysis was performed (Table 2). First, a univariate analysis was performed with plasma polyP levels and all the variables of interest using simple linear regression (Table 2). Then, a bivariate model was constructed with polyP as the dependent variable and all the variables that presented *p*-values ≤ 0.2 in the univariate analysis (Table 2). Linear regression with insulin, HOMA-IR and vWF were statistically significant in the univariate analysis (*p*-value < 0.05) (Table 2 and Appendix A). GPT presented a significant linear relationship with polyP in the bivariate model (Table 2). The adjusted model (bivariated) showed an R-squared of 0.43 and an AIC of 81.73.

### 3.2. Plasma Polyphosphate in Severe Type 2 Diabetes Adult Patients

In view of the above results in children with insulin resistance, plasma polyP was also evaluated 10 adult patients with type 2 Diabetes (Figure 3a). Patients with long-standing diabetes, dependent on insulin administration, and with cardiovascular complications were chosen (Materials and methods) and compared with 10 age-matched individuals not diagnosed with diabetes as a control group. Diabetic patients had significantly higher plasma polyP levels (Figure 3a). Likewise, diabetic patients had increased levels of von Willebrand factor (Figure 3b), as would be expected in this type of patient. Finally, platelets were separated from the blood of these patients and intraplatelet polyP levels were measured (Figure 3c). Surprisingly, polyP levels within platelets were also significantly increased in patients with type 2 diabetes (Figure 3c).

## 4. Discussion

Measurement of polyP levels in plasma has been elusive until recently when we developed an assay to determine it using its cryoprecipitate fraction [19]. In previous work, this technology enabled the study of plasma polyP levels in a healthy population, revealing a significant negative correlation between plasma polyP and the obesity stage [19].

Here, in a sample of children with overweight and obesity, we found a surprising effect opposite to that observed in healthy adults: plasma polyP levels are increased with more BMI (Figure 1a). When analyzing the biochemical parameters, we found that most of the individuals in the study had insulin resistance (Table 1).

Likewise, the group with higher BMIs presented an increase in plasma von Willebrand factor (vWF) (Figure 1b). vWF is a known indicator of inflammation. It has been recognized as an acute phase reactant [24], and its levels increases in inflammatory and metabolic disorders such as increasing insulin levels, glucose intolerance, diabetes, and obesity [23,25,26,27].

Therefore, the increase in polyP appears to be due more to the pro-inflammatory state of the patients. When we grouped the participants taking into account only the presence or absence of insulin resistance, the differences in plasma polyP levels were even more evident (Figure 2a).

Another related consequence of the metabolic syndrome is the increment of the he-patic enzymes alanine transaminase (GPT) and aspartate transaminase (GOT) in plasma [28]. The rise in GPT and GOT has been significantly associated with pre-diabetes and diabetes, making them very useful first indicators of disturbed glucose metabolism [29]. In our results, the relationship between increased plasma polyP and inflammation was evidenced by the correlation of this polymer with insulin levels (Figure 2b), as well as an increased activity of GPT (Figure 2c). In addition, a multiple linear regression analysis confirmed that polyP is statically correlated with insulin, vWF, and GPT (Table 2).

Insulin resistance has been identified as a significant cause of type 2 diabetes [4,5]. In view of the above results with insulin resistance in children, plasma polyP was also evaluated in a sample of adult patients with type 2 diabetes (Figure 3). Plasma polyP levels were also increased in adult patients with long-standing type 2 diabetes (Figure 3a). These patients have higher levels of von Willebrand factor in plasma, as expected (Figure 3b).

Previously, we had reported a correlation between platelet polyP and plasma vWF by studying healthy individuals and patients with von Willebrand disease [21]. Platelets are one of the main cellular reservoirs of polyP, which is secreted into plasma by activated platelets [11]. Therefore, we studied platelets from patients with type 2 diabetes and found that they also present higher levels of polyP (Figure 3c).

Our initial study of plasma polyP in 200 individuals showed that polyP tends to decrease in overweight and obese individuals [19]. In that sample, normal-weight, overweight, and obese groups did not have significant differences in glucose, cholesterol, triglycerides, and fibrinogen levels, among others [19], indicating that all were “metabolically healthy” regardless of their BMI. However, in this work we see that plasma polyp is increased in those patients with insulin resistance and type 2 diabetics. Preliminary results from our group show that plasma polyp also increases in gestational diabetes (Montilla and Ruiz, in preparation). Taken together, the results lead us to propose a model of the variation of plasma polyP (Figure 4). In metabolically healthy individuals, polyP is decreased in those who are overweight and obese, except when insulin resistance and diabetes have begun–when there is a pronounced increase in polyP levels (Figure 4). Then, it becomes urgent to characterize plasma polyP levels in other models of insulin resistance and diabetes, and design new studies that include large human population samples, which could definitively identify polyP as a possible biomarker in the early diagnosis of diabetes mellitus and its complications. Moreover, as a broad debate has recently begun in pursuit of the definition of “metabolic health” in obesity [30,31], further studies of the roles of polyP may serve to add it as a new identifier to differentiate metabolically healthy obesity (MHO) from metabolically unhealthy obesity (MUHO) (Figure 4).

Recently it has been reported that high extracellular phosphate increases platelet polyP content [32]. Interestingly, emerging evidence indicates that phosphate metabolism is altered in diabetes [33,34]. Hyperphosphatemia has been described in an animal model of diabetes [35], and diabetes is considered a major risk factor for hyperphosphatemia through kidney disease [36]. Hence, disturbances in the levels of plasma phosphate could be a cause for the observed higher levels of platelet and plasma polyP in diabetes.

Additionally, we observed here that the plasma of polyP is higher in healthy adults than in the children without metabolic syndrome studied (Figure 2a and Figure 3a). However, data are still preliminary to ascertain age-dependent plasma polyP differences. It would be interesting to confirm these results by increasing the number of individuals and/or conducting specific studies of polyP in plasma depending on age.

The relationship between polyP and vWF levels, which we previously described in von Willebrand disease [21], is also evident in this study for patients with insulin resistance and diabetes. Increased plasma polyP could be linked to cardiovascular complications occurring in diabetes. On the one hand, the von Willebrand factor binds to polyP, and this binding increases the binding of the factor to platelet glycoprotein Ib [21], which possibly contributes to increased platelet adhesion to the endothelium. Additionally, polyP may be causing complications through its potent effects in the plasma clotting cascade influencing hemostasis, thrombosis, and inflammation (recently reviewed in [9] and [37].

On the other hand, inositol polyphosphates (InsPs) are another type of phosphate polymers which have recently been found to have strong metabolic connections with polyP [38,39,40]. Recently, enzymes of InsPs metabolism have been related to insulin signaling [41], insulin resistance, and type 2 diabetes [42]. Therefore, it is to be expected that differences in plasma polyP levels may be connected with InsPs metabolism. This may be one of the most interesting directions for future research.

## 5. Conclusions

This work is the first demonstration of the increase in plasma polyP that occurs in insulin resistance and diabetes. As this polymer decreases in obesity in healthy people, the results suggest that this polymer could be a biomarker for the inflammatory state at the onset of diabetes. More studies are urgently needed to explore plasma polyP in several models of diabetes and inflammation.

## Figures and Tables

**Figure 1 nutrients-14-04601-f001:**
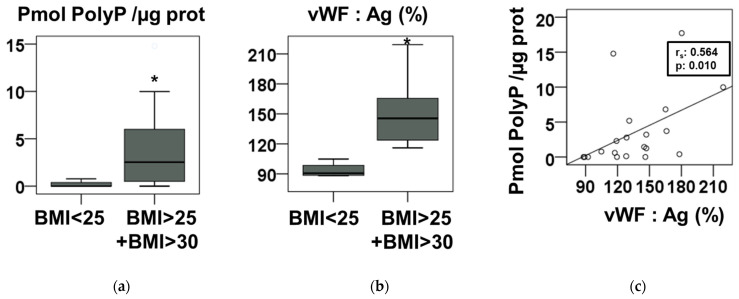
Increased blood plasma polyphosphate levels in children with BMI > 25. (**a**) Plasma polyphosphate (polyP) levels, from individuals described in Table 1, determined by a novel method that used the cryoprecipitated plasma fraction and purified recombinant yeast exopolyphosphatase (described in the Materials and methods section). PolyP concentrations are expressed in terms of phosphate. (**b**) Plasma von Willebrand factor antigen (vWF:Ag), from individuals described in Table 1, as described in the Materials and methods section. In (**a**,**b**), results are presented in a box-and whiskers plot and the asterisks indicates a statistical difference of *p*-value < 0.05, determined by Mann–Whitney test. (**c**) Graphic representation and Spearman’s correlation analysis of plasma von Willebrand factor antigen (vWF:Ag) and levels of plasma polyP of all individuals measured in (**a**,**b**). “r_s_”: Spearman’s correlation coefficient.

**Figure 2 nutrients-14-04601-f002:**
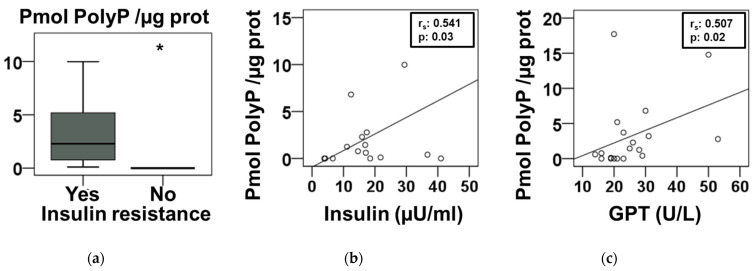
Insulin resistance in children is associated with increased blood plasma polyphosphate levels. (**a**) Plasma polyphosphate (polyP) levels, presented in Figure 1a, were reanalyzed with respect to the diagnosis of insulin resistance (IR) using HOMA-IR indexes (*n* = 16 with IR and *n* = 9 without IR). Results are presented in a box-and whiskers plot and the asterisks indicates a statistical difference of *p*-value < 0.05, determined by Mann–Whitney test. (**b**,**c**) Graphic representation and Spearman’s correlation analysis of plasma polyP and plasma insulin (**b**), and plasma alanine transaminase activity (GPT) (**c**). “r_s_”: Spearman’s correlation coefficient.

**Figure 3 nutrients-14-04601-f003:**
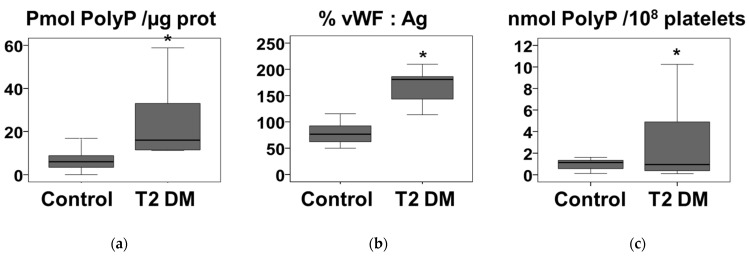
Increased plasma and platelet polyphosphate levels in severe type 2 diabetes. (**a**) Plasma polyphosphate (polyP) levels, (**b**) Plasma von Willebrand factor antigen (vWF:Ag), and (**c**) Platelet polyP levels were determined in 10 patients of type 2 diabetes mellitus (T2 DM) and 10 controls (described in the Materials and methods section). Results are presented in box-and whiskers plots and the asterisks indicate a statistical difference of *p*-value < 0.05, determined by Mann–Whitney test. Concentrations of polyP are expressed in terms of phosphate.

**Figure 4 nutrients-14-04601-f004:**
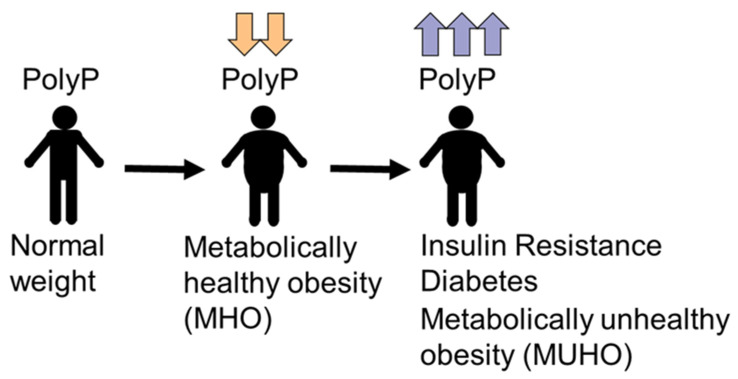
Proposed model of the variations in plasma polyphosphate in obesity, insulin resistance, and diabetes. In metabolically healthy individuals, plasma polyP tends to decrease with overweight and obesity. With the onset of insulin resistance and diabetes, polyP increases because individuals became metabolically unhealthy.

**Table 1 nutrients-14-04601-t001:** Characteristics (means ± SD) of the 25 children studied.

	BMI < 25 (*n* = 5)	BMI > 25 (*n* = 10)	BMI > 30 (*n* = 10)
Age (years)	7.4 ± 0.2	9.6 ± 3.0	12.2 ± 1.4 ^a^
Male/Female	2/3	5/5	6/4
BMI (kg/m^2^)	20.7 ± 4.8	28.4 ± 1.6 ^a^	32.37 ± 3.1 ^a,b^
TC (mg/dL)	145.5 ± 34.7	154.5 ± 20.3	149.4 ± 43.6
TG (mg/dL)	78.2 ± 42.3	94.6 ± 40.3	95.0 ± 42.8
LDL-C (mg/dL)	67.3 ± 37.3	85.1 ± 21.6	83.9 ± 39.5
HDL-C (mg/dL)	64.7 ± 8.9	50.6 ± 11.6	46.5 ± 9.3
Glucose (mg/dL)	83.2 ± 11.6	85.9 ± 4.3	93.6 ± 8.0
Insulin (µu/dL)	8.0 ± 6.4	**21.4 ± 10.2 ^a,c^**	**27.0 ± 15.9 ^a,c^**
HOMA-IR	1.7 ± 1.4	**4.4 ± 1.9 ^a,c^**	**6.2 ± 3.6 ^a,c^**
GOT (U/L)	33.0 ± 8.6	21.7 ± 4.5	23.3 ± 4.4
GPT (U/L)	29.0 ± 16.0	20.1 ± 5.7	28.4 ± 9.7

Abbreviations: BMI, body mass index; TC, total cholesterol; TG, triglycerides; LDL-C, low-density lipoprotein cholesterol; HDL-C, high-density lipoprotein cholesterol; HOMA-IR, homeostasis model assessment of insulin resistance; GOT, aspartate transaminase; GPT, alanine transaminase. ^a^ Significant difference compared with “Normal”; ^b^ Significant difference compared with “Overweight” (*p*-value < 0.05, Mann–Whitney U test). ^c^ Significant differences (*p*-value < 0.05), determined by Kruskal-Wallis test. Simultaneous significant differences ^a^ and ^c^ are indicated in bold font.

**Table 2 nutrients-14-04601-t002:** Multivariate analysis with the characteristics of the children studied.

	Univariate Analysis		Bivariate Analysis	
	Beta	*p*-Value	Beta	*p*-Value
Age (years)	0.318	0.46	-	-
Male/Female	2.481	0.2	−1.72077	0.406
BMI (kg/m^2^)	0.166	0.47	-	-
TC (mg/dL)	0.004	0.904	-	-
TG (mg/dL)	−0.023	0.43	-	-
LDL-C (mg/dL)	−0.0007	0.981	-	-
HDL-C (mg/dL)	0.081	0.349	-	-
Glucose (mg/dL)	0.0181	0.9	-	-
Insulin (µu/dL)	**0.1765**	**0.03**	0.245938	0.567
HOMA-IR	**0.7847**	**0.0325**	−0.60355	0.7629
GOT (U/L)	0.1787	0.315	-	-
GPT (U/L)	0.18	0.11	**0.242983**	**0.0167**
vWF:Ag (%)	**0.0719**	**0.03**	−0.005027	0.8958
				**R^2^ = 0.43**

Abbreviations: BMI, body mass index; TC, total cholesterol; TG, triglycerides; LDL-C, low-density lipoprotein cholesterol; HDL-C, high-density lipoprotein cholesterol; HOMA-IR, homeostasis model assessment of insulin resistance; GOT, aspartate transaminase; GPT, alanine transaminase; vWF:Ag, Plasma von Willebrand factor antigen. Significant differences (*p*-value < 0.05) are indicated in bold font.

## Data Availability

Not applicable.

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
