# Peer review of "Proinflammatory Polyphosphate Increases in Plasma of Obese Children with Insulin Resistance and Adults with Severe Type 2 Diabetes"

_nutrients, 2022, doi:10.3390/nu14214601_

Round 1
Reviewer 1 Report
1. The authors may consider changing the title, as the study population includes not only children but also adults with severe type 2 diabetes.
2. Please include the number of participants and study design (whether it’s cross-sectional, or cohort) in the abstract.
3. Line 45, since “then”, not “them”. Please correct it.
4. I’m concerned about the study population. The included 20 children with overweight/obesity were recruited from the hospital, therefore, the reason for admission would be important. Usually, children with just overweight/obesity, without other diseases are not necessarily to be in-patients. Besides, medical history is also of concern in the same regard.
5. Same concern for the adult participants.
6. In the statistical analysis, the authors should consider potential confounders. Multivariate regressions would be added except for the bivariate analysis, and state in detail the covariates adjusted in the model.
7. For the linear regression analyses conducted in Figure 1 and Figure 2, what was included in the model?
Author Response
Editorial Board Nutrients
Manuscript ID: nutrients-1990940
October 19th, 2022
We would like to thank the referees for their constructive and useful comments. After going through additional analysis and re-drafting of the paper, we feel confident that we have now clarified all their concerns.
In addition, during the review of the manuscript, we detect a minor error in Fig 2c. The data showed before correspond to the enzyme GPT, instead of GOT. We have changed the statement in the updated manuscript accordingly. This change does not alter any of the conclusions of our study. We apologize to the editor and reviewers for the mistake.
Please find below our point-by-point answers to all the comments. Our responses are in italics to make the text easier to read. Changes made in the new manuscript are highlighted using the “Track Changes” function of MS Word.
We hope that the manuscript is now acceptable for publication in Nutrients.
Reviewer 1
- The authors may consider changing the title, as the study population includes not only children but also adults with severe type 2 diabetes.
In the revised manuscript, we have changed the title to: “Proinflammatory Polyphosphate increases in plasma of Obese Children with Insulin Resistance and Adults with Severe Type 2 Diabetes”.
- Please include the number of participants and study design (whether it’s cross-sectional, or cohort) in the abstract.
Participant number and study design were included in the abstract. In addition, minor changes were needed to adjust the new abstract to 200 words.
- Line 45, since “then”, not “them”. Please correct it.
We changed the typo in the revised manuscript.
- I’m concerned about the study population. The included 20 children with overweight/obesity were recruited from the hospital, therefore, the reason for admission would be important. Usually, children with just overweight/obesity, without other diseases are not necessarily to be in-patients. Besides, medical history is also of concern in the same regard.
We thank the reviewer for this excellent observation and apologize for the misunderstanding.
Participating children with obesity or overweight were recruited from the Pediatric Endocrinology Outpatient Clinic of Puerta del Mar University Hospital, where their primary care providers had referred them due to their weight excess. Only children in which organic, syndromic and monogenic obesity was ruled out, were finally included in the study.
In the revised manuscript we added this explanatory information (Material and Methods section, 2.1.1).
- Same concern for the adult participants.
Type 2 Diabetes adults were patients of the Endocrinology Unit at the Puerta del Mar University Hospital. Inclusion criteria was diagnoses of long-term diabetes with insulin dependence and complications, such as diabetic foot or ischemic heart disease. In contrast, the controls were healthy volunteers recruited in our Research Unit at the Puerta del Mar University Hospital. The exclusion criteria for the control individuals was any medical history of Diabetes or Metabolic disorders. Inclusion criteria for the controls was matched by the age and sex with the diabetic patients.
In the revised manuscript, we added this explanatory information (Material and Methods section, 2.1.2).
- In the statistical analysis, the authors should consider potential confounders. Multivariate regressions would be added except for the bivariate analysis, and state in detail the covariates adjusted in the model.
Following the reviewer’s suggestion, a new analysis was added in the revised manuscript (New Table 2): First, a univariate analysis was performed with plasma polyP levels and all the variables of interest using simple linear regression (Table 2). Then, a bivariate model was constructed with polyP as the dependent variable and all the variables that presented p-values ≤ 0.2 in the univariate analysis (Table 2).
- For the linear regression analyses conducted in Figure 1 and Figure 2, what was included in the model?
We thank the reviewer for pointing this out. Figure 1 and Figure 2 show data from Spearman's correlation analysis (not linear regression analyses). We have changed the statement in the updated manuscript accordingly.
Respectfully,
Felix A. Ruiz, Ph.D.
Universidad de Cádiz and Hospital Universitario Puerta de Mar
Unidad de Investigación. 9ª planta
Avda. Ana de Viya, 21 11009-Cadiz, SPAIN
Phone: +34-690395217
e-mail: felix.ruiz@uca.es

Reviewer 2 Report
The manuscript entitled “Proinflammatory Polyphosphate increases in plasma of Obese 2
Children with Insulin Resistance and in Type 2 Diabetes” by Montilla et al is interesting, however there is a major concern regarding the trial.
In the title author mentioned about children with insulin resistance and type 2 diabetes but in the material and method section authors also conducted the trial on the adults as well. How they can corelate the outcomes of the adults with the children. The introduction section should be improved, authors should discuss the current hypothesis with more literature.
The linear regression between plasma von Willebrand factor antigen 149 (vWF:Ag) and levels of plasma polyP is low (r=0.584), please justify.
Please make the coherence and try to compare the study outcomes with more literature.
In the present form it is not suitable for the publication
Author Response
Editorial Board Nutrients
Manuscript ID: nutrients-1990940
October 19th, 2022
We would like to thank the referees for their constructive and useful comments. After going through additional analysis and re-drafting of the paper, we feel confident that we have now clarified all their concerns.
In addition, during the review of the manuscript, we detect a minor error in Fig 2c. The data showed before correspond to the enzyme GPT, instead of GOT. We have changed the statement in the updated manuscript accordingly. This change does not alter any of the conclusions of our study. We apologize to the editor and reviewers for the mistake.
Please find below our point-by-point answers to all the comments. Our responses are in italics to make the text easier to read. Changes made in the new manuscript are highlighted using the “Track Changes” function of MS Word.
We hope that the manuscript is now acceptable for publication in Nutrients.
Reviewer 2
- “…In the title author mentioned about children with insulin resistance and type 2 diabetes but in the material and method section authors also conducted the trial on the adults as well.”
In the revised manuscript, we have changed the title to: “Proinflammatory Polyphosphate increases in plasma of Obese Children with Insulin Resistance and Adults with Severe Type 2 Diabetes”.
- “How they can correlate the outcomes of the adults with the children.”
This is a great observation. In the revised manuscript, we discuss the results of healthy adults with children.
“We observed that the plasma of polyP is higher in healthy adults than in the children without metabolic syndrome studied (Figures 2A and 3A). However, data are still preliminary to ascertain age-dependent plasma polyp differences. It would be interesting to confirm these results by increasing the number of individuals and/or doing specific studies of polyP in plasma depending on age.”
- “The introduction section should be improved, authors should discuss the current hypothesis with more literature.”
We thank the reviewer for this suggestion. We improved the introduction by adding six more references in the revised manuscript. We feel that the changes added more coherence to the hypothesis pursued in our study.
In addition, we added more information in the discussion section that explains better the background related to the elements that correlate with plasma polyP (vWF and GPT).
“The linear regression between plasma von Willebrand factor antigen 149 (vWF:Ag) and levels of plasma polyP is low (r=0.584), please justify.”
We thank the reviewer for pointing this out. Figure 1 and Figure 2 show data from Spearman's correlation analysis (not linear regression analyses). We have changed the statement in the updated manuscript accordingly. PolyP and vWF levels showed a moderate correlation between them.
In addition, we added linear regression analysis in the revised manuscript that was statistically significant for PolyP and vWF (New Table 2).
“Please make the coherence and try to compare the study outcomes with more literature.”
We acknowledge all the observations. We believe that changes made stronger the manuscript.
Respectfully,
Felix A. Ruiz, Ph.D.
Universidad de Cádiz and Hospital Universitario Puerta de Mar
Unidad de Investigación. 9ª planta
Avda. Ana de Viya, 21 11009-Cadiz, SPAIN
Phone: +34-690395217
e-mail: felix.ruiz@uca.es

Round 2
Reviewer 2 Report
The paper is now acceptable in the present form
Author Response
We thank the reviewer for indicating that "the paper is now acceptable in its present form".